# Evidence of the Beneficial Impact of Three Probiotic-Based Food Supplements on the Composition and Metabolic Activity of the Intestinal Microbiota in Healthy Individuals: An Ex Vivo Study

**DOI:** 10.3390/nu15245077

**Published:** 2023-12-12

**Authors:** María Carmen Sánchez, Ana Herráiz, Sindy Tigre, Arancha Llama-Palacios, Marta Hernández, María José Ciudad, Luis Collado

**Affiliations:** 1Department of Medicine, Faculty of Medicine, University Complutense, 28040 Madrid, Spain; mariasan@ucm.es (M.C.S.); anaherra11@ucm.es (A.H.); stigre@ucm.es (S.T.); mallamap@ucm.es (A.L.-P.); lcollado@ucm.es (L.C.); 2GINTRAMIS Research Group (Translational Research Group on Microbiota and Health), Faculty of Medicine, University Complutense, 28040 Madrid, Spain; 3CAPSA FOODS, 33199 Granda-Siero, Asturias, Spain; marta.hernandez@capsa.es

**Keywords:** probiotic-based food supplements, *Bifidobacterium longum* ES1, *Lactobacillus acidophilus* NCFM^®^, HOWARU^®^ restore, short-chain fatty acids (SCFA), branched short-chain fatty acids (BCFA), ammonium, lactate, *Lactobacillus* spp., *Bifidobacterium* spp.

## Abstract

Scientific evidence has increasingly supported the beneficial effects of probiotic-based food supplements on human intestinal health. This ex vivo study investigated the effects on the composition and metabolic activity of the intestinal microbiota of three probiotic-based food supplements, containing, respectively, (1) *Bifidobacterium longum* ES1, (2) *Lactobacillus acidophilus* NCFM^®^, and (3) a combination of *L. acidophilus* NCFM^®^, *Lactobacillus paracasei* Lpc-37™, *Bifidobacterium lactis* Bi-07™, and *Bifidobacterium lactis* Bl-04™. This study employed fecal samples from six healthy donors, inoculated in a Colon-on-a-plate^®^ system. After 48 h of exposure or non-exposure to the food supplements, the effects were measured on the overall microbial fermentation (pH), changes in microbial metabolic activity through the production of short-chain and branched-chain fatty acids (SCFAs and BCFAs), ammonium, lactate, and microbial composition. The strongest effect on the fermentation process was observed for the combined formulation probiotics, characterized by the significant stimulation of butyrate production, a significant reduction in BCFAs and ammonium in all donors, and a significant stimulatory effect on bifidobacteria and lactobacilli growth. Our findings suggest that the combined formulation probiotics significantly impact the intestinal microbiome of the healthy individuals, showing changes in metabolic activity and microbial abundance as the health benefit endpoint.

## 1. Introduction

The biological system that involves the gut microbiome contains a large diversity of microorganisms and their genomes, with a predominance of bacteria and their coding genes that produce thousands of metabolites, playing a fundamental role in maintaining intestinal homeostasis [1,2]. In a healthy adult, this ecosystem is dominated by bacteria, mainly the *Firmicutes* and *Bacteroidetes* phyla, followed by *Proteobacteria*, *Actinobacteria*, and *Verrucomicrobia*, which constitute the so-called “core microbiome” but are subject to continuous variations in their relative abundance and richness, making it a unique ecosystem [3,4]. The importance of the gut microbiome lies in its intimate relationship with the individual’s health, both at the level of the intestinal mucosa and the immune system. Interactions within the host–gut microbiota promote the functional and structural maturation of the gastrointestinal tract, acting mainly on surface maturation and peristalsis, maintaining the integrity of the intestinal epithelial barrier by maintaining cell–cell junctions, and promoting epithelial repair after injury. At the same time, the microbiota provides a physical barrier against incoming pathogens through competitive exclusion by occupying binding sites, consuming nutrient sources, and producing antimicrobial substances, and also by stimulating the host to produce various antimicrobial compounds. Additionally, the metabolic processes of the intestinal microbiota benefit the host and participate in acquiring nutrients, producing essential human metabolites such as vitamins, and processing xenobiotics. The gut microbiota provides capabilities for the fermentation of nondigestible substrates such as dietary fibers and endogenous intestinal mucus. This fermentation supports the growth of short-chain fatty acids (SCFA)-producing bacteria, which are the main source of energy for human colonocytes. Fermentation also supports the grown of microorganisms that facilitate apoptosis in colon cancer cells and that activate intestinal glucose and energy homeostasis [5,6,7]. On the other hand, evidence indicating that an increase in the *Firmicutes*/*Bacteroidetes* ratio is associated with obesity and that a certain intestinal microbiota dysbiosis (imbalance in the composition and function of intestinal microbiome) correlates with several diseases, ranging from localized gastrointestinal disorders to extraintestinal conditions, including neurological, respiratory, metabolic, hepatic, and cardiovascular diseases [8]. Due to these associations, significant scientific effort is being focused on studying the possible causality and mechanisms related to disease mediated by the intestinal microbiota, searching for new therapeutic and preventive strategies [8,9]. Numerous scientific studies have highlighted the major influence that diet and the consumption of specific health-related foods, such as probiotics, prebiotics, and symbiotics, have on the composition of the gut microbiota and their balance and thus on human health through the effects on intestinal homeostasis and inflammation inhibition [10,11].

Probiotics, defined as living microorganisms that confer a health benefit to the host when administered in appropriate amounts [12], can directly modulate gut health at the gastrointestinal level through the production of health-promoting metabolites and can indirectly modulate it by generating metabolites that can be employed by other colonic bacteria as cross-feeding substrates to produce health-promoting metabolites. The direct or indirect metabolic action of probiotics includes catalytic pathways for the metabolism of complex carbohydrates that produce the SCFAs. Probiotics also modulate cytokines related to inflammation and have an influence on anti-proliferative lipids, which represent an essential energy source for gastrointestinal epithelial cells, microbial-derived bioactive metabolites such as vitamins K and B, hormones, and neurological signaling molecules, among others [7,13,14]. Thus, by modulating the activity and composition of the colonic microbiota, the probiotic can indirectly affect the host, a process commonly referred to as the host–microbiota interaction. This interaction suggests a contribution toward strengthening the activity of the intestinal mucosal barrier, in particular influencing intestinal epithelial cells and immune cells, essential for preventing and treating intestinal diseases such as diarrhea (acute, traveler’s, antibiotic-induced, and rotavirus-induced), inflammatory bowel diseases (Crohn’s disease, pouchitis, or ulcerative colitis due to *Clostridium difficile*), irritable bowel syndrome, and necrotizing enterocolitis, among others [15,16,17,18]. Recent studies have suggested that emerging next-generation probiotics have an anticancer effect and can help control insulin resistance as a treatment for type 2 diabetes, reverse obesity, and protect against intestinal disease [12,19].

Due to the numerous beneficial effects provided by probiotics, a deep knowledge is required of the action and underlying mechanisms of the action of probiotics on human microbiota, intestinal health, and general health [20,21]. Lactobacilli, along with *Bifidobacterium* species, are historically considered the most common probiotics that, in controlled studies in humans, have demonstrated health benefits. The yeast *Saccharomyces boulardii* and certain species of *Escherichia coli* and *Bacillus* have also been used. Newcomers to the probiotic ranks include *Clostridium butyricum*, recently approved as a novel food in the European Union (World Gastroenterology Organization Global Guidelines for Probiotics and Prebiotics; www.worldgastroenterology.org, accessed on 1 February 2023). In this context, only a few studies have explored how probiotics affect the metabolism and composition of a healthy intestinal microbiota to maintain a balanced environment. Most studies mainly focus on alleviating symptoms associated with disorders caused directly or indirectly by an imbalance in the gut microbiota. In fact, to improve their effectiveness against symptoms related to these disorders, commercial probiotics are often formulated with a combination of multiple probiotic strains. Alternatively, they might contain a single strain of probiotics or a mixture, combined with natural bioactive agents such as essential oils, coumarins, tannins, flavonoids, phytosterols, amino acids, omega-3 fatty acids, or proteolytic enzymes. These components help relieve symptoms such as abdominal pain, bloating, irregular bowel movements, reflux, and heartburn. It is important to note that these ingredients not only influence intestinal cells by modulating inflammation, redox state, pain perception, and immune responses, but they also potentially affect the balance and activity of the intestinal microbiota [22,23,24,25,26].

Therefore, the aim of this ex vivo study was to evaluate how probiotic-based food supplements containing *Lactobacillus* and *Bifidobacterium* strains impact the composition and metabolic activity of the intestinal microbiota [12]. Three probiotic-based food supplements were selected due to their success in treating intestinal dysfunction: one with *Bifidobacterium longum* ES1 combined with lavender extract, glutamine, and omega-3 fatty acids for symptoms such as abdominal discomfort; another with *Lactobacillus acidophilus* NCFM^®^ with ingredients such as lactase, bromelain, chamomile, fennel, and mint for acid reflux conditions; and a third combining *L. acidophilus* NCFM^®^, *Lactobacillus paracasei* Lpc-37™, *Bifidobacterium lactis* Bi-07™, and *Bifidobacterium lactis* Bl-04™ to address intestinal stress. To undertake this study, fecal samples from six healthy donors were used and were inoculated in a Colon-on-a-plate^®^ system, which represents a high-throughput biorelevant ex vivo simulation of the colon’s physiology and microbiology [27]. The system has been optimized to perform short-term colonic simulations of up to 48 h, studying the changes in the composition of the microbial community and the variations in metabolite production.

## 2. Materials and Methods

### 2.1. Probiotic-Based Food Supplements

The following commercial formulations for Capsa Foods (CAPSA FOODS, Granda-Siero, Asturias, Spain) were employed in the study: (a) MultiPro (MP), containing an exclusive combination known as HOWARU^®^ Restore, combining *L. acidophilus* NCFM^®^, *L. paracasei* Lpc-37™, *B. lactis* Bi-07™, and *B. lactis* Bl-04™, to treat episodes of intestinal stress; (b) GastricPro (GP), a food supplement that combines the specific probiotic strain *L. acidophilus* NCFM^®^ with active ingredients such as lactase, bromelain, and extracts of chamomile, fennel, and mint, which are used for acid and reflux conditions, flatulence, gas, and drug-induced reflux and acidity; and (c) ProIntestino (PI), a food supplement that combines the specific probiotic strain *B. longum* ES1 with active ingredients such as lavender flower extract (which improves digestive comfort in cases of stress), glutamine, and omega 3 fatty acids (EPA + DHA), which are appropriate for symptoms and signs of abdominal pain, swelling, and distension and evacuation abnormalities.

### 2.2. Collection and Preservation of Fecal Inocula from Healthy Donors

The intestinal microbiota samples employed in the ex vivo model were obtained from fecal inocula from the 6 healthy donors who had no history of antibiotic use during the 3 months prior to the experiment (age range 20–40 years; 3 participants were men and 3 were women; Biobank EC approval number ONZ-2022-0267). The fecal samples were collected according to the protocol approved by the ethics committee of the University Hospital Ghent (reference number B670201836585). The donors’ informed consent or that of their legal representatives was obtained. To manage the simultaneous inoculation of the healthy donors’ fecal microbiota required for the Colon-on-a-Plate (CoaP^®^) assay, the fecal samples were cryopreserved prior to the experiment. In short, fecal suspensions were prepared and mixed with an in-house optimized cryoprotectant, i.e., a modified version of the cryoprotectant developed by Hoefman et al. [28]. The obtained suspensions were flash frozen and then preserved at −80 °C (cryostock) until use. Just before the experiment, an aliquot was defrosted and immediately added to the reactors.

### 2.3. Colon-on-a-Plate™ Model

The CoaP^®^ fermentation model (a well-suited ex vivo model) employed for this study is based on deep-well plates, a format that ensures identical physical conditions across the experiment, thereby ensuring reproducibility [27]. Briefly, at the start of the experiment, the wells (on 24-well plates with 10.4 mL of volume capacity; Thomson, Oceanside, Canada) were filled with a fiber-supplemented background nutritional medium–fecal inoculum blend (nutritional medium PD02; ProDigest, Ghent, Belgium), representative of the colon environment, containing basal nutrients of the colon, including fiber, peptone, yeast extract, mucin, and L-cysteine. Probiotic test supplements (final concentration 1 × 10^7^ CFU/mL) or blank medium (for negative control) were added to the respective wells. Lastly, 10% (*v*/*v*) of a cryopreserved fecal inoculum containing 75 g/L of fecal material from a healthy human donor was added to each well, serving as the microbial source. The total volume in the wells was 7 mL. In each step, anaerobiosis was ensured by conducting the experiment in an anaerobic chamber where oxygen levels were carefully monitored. The plates were incubated in an anaerobic atmosphere at 37 °C for 48 h, under continuous mild shaking (90 rpm). Each condition was tested individually, resulting in 24 wells (3 treatments and 1 negative control in 6 donors). 

### 2.4. Study Endpoints

Samples were collected 48 h after the start of the experiment. Assessments of the microbial activity (pH, short-chain fatty acids (SCFA), branched short-chain fatty acids (BCFA), lactate and ammonium production) and of the community composition were performed.

#### 2.4.1. Microbial Metabolomics Activity Analysis

The pH of the wells, determined by the production of microbial metabolites, was measured using a Senseline F410 pH meter (ProSense, Oosterhout, The Netherlands). The SCFAs (butyrate, propionate, and acetate) and BCFAs (isobutyrate, isovalerate, and isocaproate) were quantitatively assessed using the methods described by De Weirdt et al. [29] and De Boever et al. [30]. Lactate concentrations were determined using the Enzytec™ kit (R-Biopharm: Darmstadt, Germany), according to the manufacturer’s instructions. Ammonium levels were determined according to the colorimetric analysis method described by Van de Wiele et al. [31], using the indophenol blue spectrophotometric method.

#### 2.4.2. Microbial Community Analysis

The abundance of the 2 dominant bacterial phyla (*Firmicutes* and *Bacteroidetes*) were monitored using quantitative polymerase chain reaction (qPCR), as well as the 3 specific groups that are typically of interest, namely *Bifidobacterium*, *Lactobacillus*, and *Akkermansia muciniphila* (a member of the *Verrucomicrobia* phylum). The qPCR protocols for the *Firmicutes* and *Bacteroidetes* phyla were previously described by Guo et al. [32]; the qPCR protocol for *A. muciniphila* was performed as reported in Collado et al. [33]; and the qPCR protocols for quantifying *Lactobacillus* spp. and *Bifidobacterium* spp. were conducted as described by Furet et al. [34] and Rinttilä et al. [35], respectively.

### 2.5. Statistical Methods

To evaluate the statistical significance of the changes in metabolite production (treatment-induced), we performed paired 2-sided *t*-tests in which each donor was considered a replicate measurement (thereby resulting in 6 “replicate” measurements per treatment). An effect was considered statistically significant if the *p*-value was <0.05. Statistical significance was therefore reached if an effect was consistent across most of the donors.

For the qPCR results (per donor and per bacterial group), we calculated the fold changes (ratio of the abundance in treatment (T) versus the abundance in blank (B), i.e., (T/B)). To present the difference in abundance of the various bacterial groups in treatment versus blank in the volcano plots, we calculated the log2 of the mean fold change (T/B) across donors (log2 (mean (T/B))). A positive number indicates stronger enrichment in treatment than in blank; a negative value indicates lower enrichment in treatment than in blank. *p*-values were calculated for each bacterial group based on the log-transformed bacterial abundance in treatment and in blank, using the 6 donors as “replicates”. A QQ-plot indicated that the data were normally distributed upon log-transformation, justifying the use of a paired Student’s *t*-test to evaluate the statistical significance of the treatment effect.

The statistical analyses were performed using GraphPad Prism version 9.3.1 for Windows (GraphPad Software, San Diego, CA, USA). 

## 3. Results

### 3.1. Overall Fermentative Activity and Changes in Metabolite Production

The changes in pH during the colonic incubation provided a good indication of overall fermentation. Figure 1a shows the pH measurements at 48 h, which indicated that there were no excessive pH decreases (the lowest pH across all conditions was 5.91). There were no significant changes in pH for any of the probiotic exposures compared with the negative control (*p* = 0.716 for MP, *p* = 0.363 for GP, and *p* = 0.115 for PI). The pH results for the individual donors are shown in Appendix A.

SCFA (butyrate, propionate, and acetate) levels, which represent carbohydrate metabolism in the colon, were measured at 48 h. The SCFA levels with probiotic supplementation versus blank are shown in Figure 1b–d. MP (combining *L. acidophilus* NCFM^®^, *L. paracasei* Lpc-37™, *B. lactis* Bi-07™, and *B. lactis* Bl-04™) and PI (*B. longum* ES1) did not affect acetate production (*p* = 0.459 and *p* = 0.082, respectively). In contrast, GP (*L. acidophilus* NCFM^®^) treatment was associated with lower acetate concentrations than the negative control of 5%, resulting from a lower production in 5 of the 6 donors, a statistically significant result (*p* = 0.039). Propionate production was not affected by any of the probiotics (the concentrations were similar to those of the control) (*p* > 0.05 in all cases). In terms of butyrate quantification, the MP condition yielded more butyrate than the blank in 5 of the 6 donors (23% increase), which was statistically significant (*p* = 0.007), whereas GP and PI did not affect butyrate production (*p* > 0.05 in both cases). The SCFA results for the individual donors are shown in Appendix A.

We also measured the levels of ammonium (NH_4_^+^) and less abundant fatty acids, the BCFAs (isobutyrate, isovalerate, and isocaproate), as markers of protein metabolism. The results are shown in Figure 1e,f and indicate that the MP condition was characterized by reduced concentrations of proteolytic markers BCFA and NH_4_^+^ in each donor, reaching statistical significance (*p* = 0.024 and *p* = 0.002, respectively). The mean reduction across all donors was 16% for BCFAs and 5% for NH_4_^+^. Neither GP nor PI supplementation significantly altered the production of proteolytic markers (*p* > 0.05 in both cases). The results for BCFAs and NH_4_^+^ for the individual donors are shown in Appendix A.

Ultimately, lactate was not detected at 48 h of incubation. Given that lactate is a typical intermediate product, this result does not necessarily imply that lactate was not produced during the incubation but might have been produced during the early stages of the incubation to be efficiently consumed at 48 h to produce propionate and/or butyrate (lactate accumulation did not occur). 

### 3.2. Changes in Microbial Community Composition

We analyzed the effects of the three probiotic-based food supplements on the microbial abundance across all donors. For *Lactobacillus* and *Bifidobacterium* spp., which are considered health-related lactic acid bacteria, the results showed that the abundance of *Bifidobacterium* was not affected by GP or PI. In contrast, *Bifidobacterium* levels were significantly increased after treatment with MP, likely due to the enrichment of intestinal bifidobacteria or the enrichment of *Bifidobacterium* spp. in the probiotic supplement (*p* < 0.001) (Figure 2). The abundance of *Lactobacillus* moderately increased upon GP treatment (*p* = 0.066) and significantly increased upon MP treatment (*p* = 0.019). As for bifidobacteria, the effect varied between the donors, suggesting that either intestinal lactobacilli or probiotic lactobacilli were enriched. PI did not alter *Lactobacillus* levels (*p* > 0.900) (Figure 2). The results for the individual donors are shown in Appendix A.

*A. muciniphila* was detected in the fecal material of four of the donors, with an abundance below the limit of quantification (1 × 10^5^ counts/mL) in the other two donors. None of the treatments affected the growth of *A. muciniphila* (*p* > 0.05 in all cases) (Figure 3), of *Bacteroidetes* spp. (one of the dominant bacterial phyla in the gut) (*p* > 0.05 in both cases) (Figure 3), or of *Firmicutes* phylum (also a dominant bacterial phylum in the human gut microbiome) (*p* > 0.05 in both cases) (Figure 3). The results for the individual donors are shown in Appendix A.

## 4. Discussion

Using an ex vivo colonic model, the present study revealed that the combination of *L. acidophilus* NCFM^®^, *L. paracasei* Lpc-37™, *B. lactis* Bi-07™, and *B. lactis* Bl-04™ caused a significant effect on the intestinal microbiome of the healthy individuals, showing changes in metabolic activity and microbial abundance as the endpoint for a health benefit, an improvement in the clinical condition, and a reduction in disease risk. We can thereby postulate that the probiotics are a powerful preventive and therapeutic alternative. Mostly minor effects were observed after exposure to *B. longum* ES1 or *L. acidophilus* NCFM^®^ supplemented with diverse active ingredients.

The development of in vitro and ex vivo models of the human colon has been promoted in recent years as a tool for studying (under controlled conditions) the effects of products to be tested on the intestinal microbiota, helping to determine the advantages of their application in human health [27,36,37,38,39]. The technology employed in the present study helped assess the mechanisms underlying the effects of test formulations on the composition and functioning of the gut microbial community, enabling the simultaneous testing of multiple probiotic formulations, thereby ensuring high reproducibility, given that the same physical conditions can be ensured for all simulated conditions [27]. The colon environment was mimicked using a representative background nutritional medium and a bacterial inoculum from a human donor as the microbial source. During the study, we verified that the production levels of the various metabolic markers in the negative control conditions corresponded to expectations. In fact, the background nutritional medium in the reactors contained fermentable nutrients (including fiber, peptone, yeast extract, mucin, and L-cysteine) that, after fermentation by intestinal bacteria, produce basic levels of metabolic markers. Furthermore, the pH profiles across all conditions indicated that the fermentation processes in the colonic simulations occurred under optimal conditions to support the growth of a broad diversity of gut microbial community members, allowing for cross-feeding interactions, should there be any. Both criteria provided a solid basis for evaluating the probiotic properties of the investigated agents.

Research in recent years on the effects of probiotics on human health have focused primarily on situations of abnormal intestinal homeostasis, i.e., in disease-related scenarios. Several clinical trials have demonstrated the anti-inflammatory effects of strains of *Bifidobacterium* during the course of various medical conditions, such as constipation, ulcerative colitis, and celiac disease [40,41,42,43]. The influence of *L. acidophilus* NCFM^®^ on the improvement of type I allergies has been studied, while a modulation of the immune system has been observed with *L. paracasei* Lpc [44,45]. The combination of the four strains, commercially known as HOWARU^®^ Restore, has demonstrated beneficial effects on gastrointestinal symptoms in patients with constipation [46] and a significant improvement in diarrhea outcomes [47]. Although these studies are heterogeneous with regard to the included strains and populations, the accumulated evidence supports the assertion that the health benefits can be measured by diverse parameters. In contrast, the present study was conducted using microbiota from healthy donors to increase the evidence of its preventive capacity. The study’s pH endpoints were measured after 48 h of exposure. The pH of the incubations is an indirect result of bacterial metabolism and can be employed to predict the effect that a probiotic exposure will have on intestinal pH. Indeed, pH is determined by the production of SCFA/BCFA/lactate/NH_4_^+^ and can therefore quickly provide insights into whether the treatment effects in terms of these endpoints can be expected. The pH values of the colonic medium in our colonic model were similar to those of the control, which ensures the maintenance of optimal conditions for bacterial growth in the colon and for fermentation processes and suggests that if there had been an increase in the formation of SCFAs or lactate, which are the products that mainly acidify the environment, this increase would be slight. Indeed, a significant increase in lactate after 48 h of incubation was not detected, and although there were significant changes in SCFA levels during exposure to certain probiotic formulations, the magnitude of these changes was not sufficient to cause a significant change in pH. 

The study assessed the gut microbial production of lactate, a product and substrate of lactic acid intestinal bacteria, which acts as an acidifying agent, lowering the pH of the medium and thereby acting as an antimicrobial [10]. However, lactate does not usually accumulate, because it is rapidly converted by the process known as cross-feeding into propionate or butyrate by other microorganisms, such as *Eubacterium hallii* and *Anaerostipes caccae* or *Megasphaera* spp. [10,48], which might be why it was not detected in our study, instead resulting in an increase in butyrate. Many of the markers that have been evaluated are products and, in turn, substrates or intermediates of other metabolic pathways; therefore, if high levels of a certain parameter, such as lactate, are not detected, it does not necessarily mean that it was not produced during incubation but rather it might have been generated during the early stages and consumed or transformed into other products such as SCFA butyrate.

We also attempted to determine the effects of probiotic exposure on saccharolytic (SCFA and lactate) and proteolytic (BCFA and ammonium) markers that can be compared with typical fermentation patterns for normal gastrointestinal microbiota. Changes in SCFA production are related to the beneficial effects because acetate and propionate act as an energy source for peripheral tissues. Acetate can be produced by numerous different gut microbes (including *Bifidobacterium* spp., *Bacteroides* spp., and *Lactobacillus* spp.), is a primary metabolite generated from substrate fermentation, and is a potential substrate for lipid synthesis. Propionate can be produced by various gut microbes (mainly *Bacteroides* spp., *A. muciniphila*, and *Veillonellaceae*) and can reduce cholesterol and fatty acid synthesis in the liver. Butyrate is mostly produced by members of the *Lachnospiraceae* and *Ruminococcaceae* families in a process called cross-feeding, converting acetate and/or lactate (along with other substrates) to the health-related butyrate, one of the most important energy sources for colonocytes, helping to renew the intestinal epithelium and showing immunosuppressive effects by regulating various cytokines and proinflammatory receptors [49,50]. BCFA (isobutyrate, 2-methylbutyrate, or isovalerate) and ammonium are compounds that indicate an increase in protein metabolism in which potentially toxic or carcinogenic components, such as *p*-cresol and *p*-phenol, are also produced and negatively affect health [1,51,52]. The most striking effects in this regard were produced in our assay by the combination of the four probiotic strains, given that at 48 h the probiotic exposure produced a significant increase in butyrate and a significant reduction in BCFA and ammonium.

These findings have been reported in previous studies for the same probiotic strains employed in the present study. For the *B. longum* ES1 strain, Zhang et al. [53] conducted a study with mice with colitis that were supplemented with the probiotic strain, reporting an increase in SCFA content. In our assay, the probiotic strain was incapable of exerting any beneficial effect that could be reflected in changes in pH, SCFA, BCFA, or NH_4_^+^ or in bacterial community composition, possibly due to the in vitro status and the short microbiome exposure time. Caviglia et al. [43], in a study of patients with irritable bowel syndrome, demonstrated the probiotic strain’s ability to modulate the composition of the intestinal microbiota, with anti-inflammatory activity that improves gastrointestinal symptoms; effects that we were unable to detect in our study.

A number of clinical studies concluded that *L. acidophilus* NCFM^®^ not only had the ability to survive the gastrointestinal passage but also had immunomodulatory and anti-inflammatory effects, improving intestinal integrity [54,55,56]. Vemuri et al. [57] conducted a trial with *L. acidophilus* in healthy elderly mice and observed an increase in the abundance of beneficial bacteria such *Lactobacillus* spp., *A. muciniphila*, and *Firmicutes*, as well as a decrease in *Bacteroidetes* and an increase in butyrate levels. Björklund et al. [58] recorded putative beneficial changes in microbiota in elderly participants supplemented with a symbiotic product consisting of the probiotics *L. acidophilus* NCFM^®^ and lactitol, with increases in the total levels of endogenous bifidobacteria and lactobacilli. Kaplan et al. [59] described the dynamics of *L. acidophilus* NCFM^®^ and observed bacterial populations that changed from rat fecal samples fed with NCFM^®^, mainly *Lactobacillus johnsonii* and *Ruminococcus flavefaciens*. Based on our in vitro observations, the combination of *L. acidophilus* NCFM^®^ and three other bacterial strains was required to obtain a significant increase in the *Bifidobacterium* and *Lactobacillus* genera, a significant increase in butyric acid, and a significant reduction in NH_4_^+^ and BCFAs. 

The *L. acidophilus* NCFM^®^ strain is usually studied together with *B. lactis* Bi-07 and *B. lactis* BI-04. The first two strains were studied by Ringel-Kulka et al. [60] in the functional disorders of the intestine, concluding that this probiotic combination reduces inflammatory symptoms and supports the role of probiotic bacteria in treating these disorders. Similarly, an experiment with weaned rats that were administered *L. acidophilus* NCFM^®^ and *B. lactis* Bi-07 for 4 weeks showed an increase in carbohydrate metabolism with the consequent generation of SCFA and adenosine triphosphate in addition to an increase in *Firmicutes*. Our study partially supports these results, showing that the use of the combination of these strains, together with *L. paracasei* Lpc-37™ and BI-04™, generated an increase in butyrate in five of the six donors and a significant increase in the genera *Bifidobacterium* and *Lactobacillus*.

The same combination of four probiotic strains (*L. acidophilus* NCFM^®^, *L. paracasei* Lpc-37™, *B. lactis* Bi-07™, and *B. lactis* BI-04™) was studied by Ouwehand et al. [56], demonstrating the dose–response relationship in patients with symptoms of diarrhea associated with antibiotics and *C. difficile*, resulting in higher doses of the probiotic supplement being more effective than lower doses, affecting the relief of symptoms and its duration. The combination of these strains, in addition to helping us to achieve a balance in the intestinal microbiota, helps maintain immune system defenses and healthy respiratory functions [61]. Similar results were observed by Wolfe et al. [62], who observed apparent taxonomic differences between treatments and timepoints. Participants administered the probiotic combination had a reduction in *Verrucomicrobiaceae* at week 8, with a significant reduction in *Bacteroides* between weeks 0 and 4 in the probiotic-treated participants. *Ruminococcus* (family *Lachnospiraceae*) tended to be more abundant at week 8 than at week 4 within the placebo group and were more abundant at week 8 than at week 0 within the probiotic group. The results are similar to those of previous studies that have associated these taxa with probiotic use and with the mitigation of *C. difficile* infection symptoms.

There is limited information on the use of probiotics to reduce the toxic metabolites of protein fermentation in humans. In a study with healthy participants, De Preter et al. [63] demonstrated that *Lactobacillus casei* and *Bifidobacterium breve* significantly decreased urinary *p*-cresol and favorably affected ammonia metabolism. In our study, the toxic metabolites produced by protein fermentation were not directly measured, but the related markers were evaluated, demonstrating that of the three products studied, only the combination of the four probiotic strains was capable of reducing BCFA and NH_4_^+^ levels, which is also associated with beneficial effects for the individual. 

Shifts in community composition were also assessed in the study and the referenced studies. Given that the human gut microbiome is composed of a small number of bacterial phyla, more specifically *Bacteroidetes*, *Firmicutes*, *Actinobacteria*, *Verrucomicrobia*, and *Proteobacteria*, the specific groups targeted in this study covered the main phyla. *Bacteroidetes* spp., one of the dominant bacterial phyla in the gut, has members (*Bacteroides*) capable of producing acetate and/or propionate and have an advanced enzymatic toolkit, enabling them to degrade complex molecules. Although *Bacteroides* is mainly associated with the digestion of dietary fiber, the genus can also incorporate amino acids, which can be used as an energy source and for maintaining cell structures. Another of the dominant bacterial phyla in this habitat is the *Firmicutes* phylum, which encompasses a collection of bacteria with highly diverse metabolite production capabilities. The phylum contains acetate-producers (e.g., *Acidaminococcaceae* and *Christensenellaceae*), lactate-producers (e.g., *Lactobacillaceae* and *Eubacteriaceae*), propionate-producers (e.g., *Veillonellaceae* and *Acidaminococcaceae*), and butyrate-producers (e.g., *Lachnospiraceae* and *Ruminococcaceae*). However, none of the treatments affected the growth of *Bacteroidetes* or *Firmicutes* spp. 

Lactobacilli and bifidobacteria, regarded as beneficial saccharolytic bacteria that can produce lactate and acetate, significantly increased their abundance after probiotic exposure, mainly due to the combination of the four probiotic strains. This study was unable to determine whether the significant increase in these bacteria was due to the incorporation of the probiotic strains into the model’s intestinal microbiome or whether they exerted their effect on species already present in the intestinal biofilm. This study also considered *A. muciniphila*, a propionate producer and mucin-degrading bacterium typically prevalent in the distal colon region, whose presence in the gut is associated with health benefits, given that inverse relationships between the colonization of *A. muciniphila* and inflammatory conditions or obesity have been observed [64]. Although the bacteria was detected in most of the donors in our study, its abundance did not vary due to the effect of the probiotic strains. 

The probiotic strains *B. longum* ES1 and *L. acidophilus* NCFM^®^ employed in this study were accompanied by bioactive ingredients. In contrast to findings reported by other authors who observed a beneficial modulation of intestinal microbiota through the use of, for instance, lavender extract, fennel, chamomile, or omega 3 fatty acids [22,23,24,25,26], our results indicate that in the two supplemented formulations, PI and GP, the additional components did not have a significant impact on the composition or metabolism of the intestinal microbiota. Lastly, the results support the pre-existing scientific evidence that the benefits increase if more than one probiotic, such as lactobacilli, is combined with bifidobacteria [65,66].

## 5. Conclusions

In conclusion, the results of our study showed that overall, the impact of the tested probiotic agents on the fermentation process of the healthy human microbiome is strain-dependent. The combination of the probiotic strains *L. acidophilus* NCFM^®^, *L. paracasei* Lpc-37™, *B. lactis* Bi-07™, and *B. lactis* BI-04™ increased the levels of *Bifidobacterium*, *Lactobacillus*, and butyrate and reduced the production of BCFA and NH_4_^+^ (*p* < 0.05 in all cases), highlighting the importance of this combination of strains for future research in this area, human health, and clinical practice. There was a significant effect on the intestinal microbiome of the healthy individuals, observing changes in metabolic activity and microbial abundance, which could constitute a reduction in disease risk. We can thereby postulate that the use of probiotics could be a powerful preventive approach. 

## Figures and Tables

**Figure 1 nutrients-15-05077-f001:**
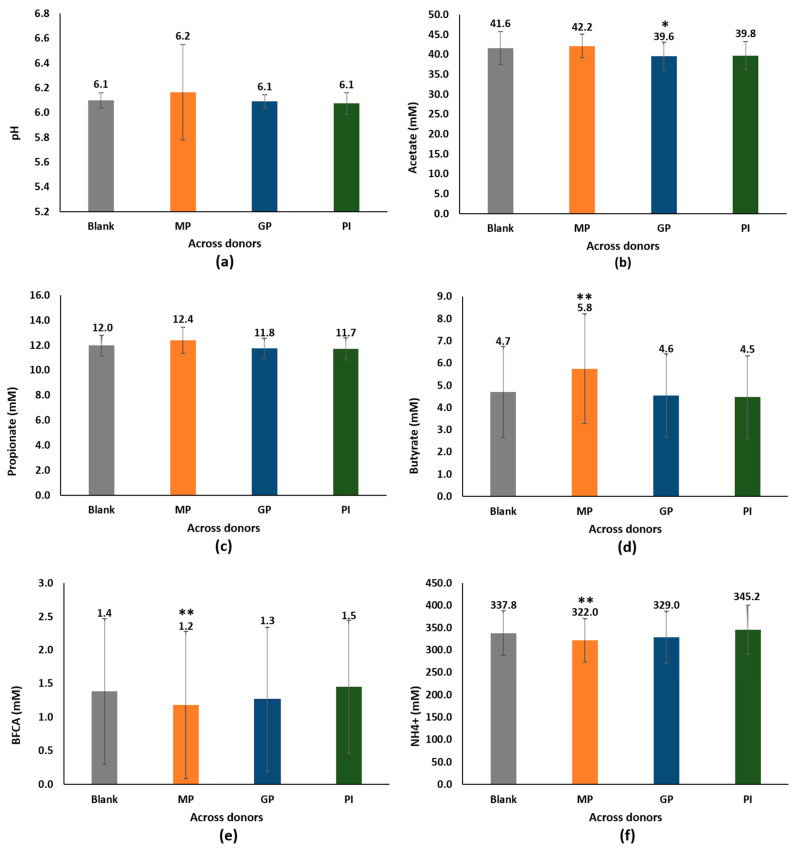
Effect of three probiotics-based food supplements on the metabolomics activity of the intestinal microbiota in healthy adult donors, in reference to (**a**) pH, (**b**) acetate, (**c**) propionate, (**d**) butyrate, (**e**) BFCAs, and (**f**) ammonium, in the ex vivo model CoaP^®^, after 48 h starting the exposition. * *p* < 0.05, ** *p* < 0.03 for differences between the supplemented and blank samples. Data are plotted as mean (all six donors) ± standard error. MP: MultiPro, containing a combination of *L. acidophilus* NCFM^®^, *L. paracasei* Lpc-37™, *B. lactis* Bi-07™, and *B. lactis* Bl-04™ strains; GP: GastricPro, containoing *L. acidophilus* NCFM^®^, with active ingredients such as lactase, bromelain, and extracts of chamomile, fennel, and mint. PI: ProIntestino, which combines *B. longum* ES1 and active ingredients such as lavender flower extract, which improves digestive comfort in cases of stress, glutamine, and omega 3 fatty acids (EPA + DHA).

**Figure 2 nutrients-15-05077-f002:**
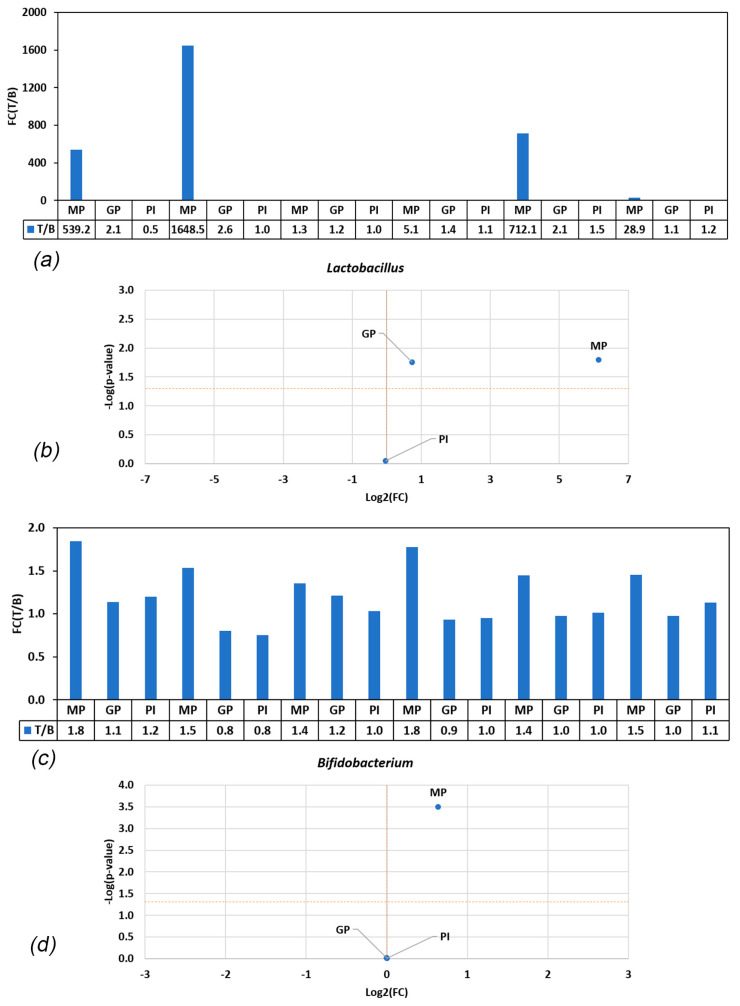
(**a**,**c**) Ratios of *Lactobacillus* spp. and *Bifidobacterium* spp. abundances in treatment versus blank (T/B) in the various conditions. Values > 1 indicate enrichment in treatment; values < 1 indicate enrichment in blank. FC = fold change; T = treatment; B = blank. (**b**,**d**) Volcano plot showing difference in abundance of *Bifidobacterium* and *Lactobacillus* spp. in blank and treatment. Statistical significance (−log(*p*-value)) is plotted in the function of fold change (treatment/blank), classifying treatments into one of four categories: non-significantly lower enrichment than blank control (**bottom left**), significantly lower enrichment than blank control (**top left**), non-significantly higher enrichment than blank control (**bottom right**), and a significantly higher enrichment than blank control (**top right**). The horizontal orange line indicates the level above which an effect is statistically significant compared to the blank across the six donors. The vertical orange line marks the separation between stronger enrichment versus blank control (**right**) or lower enrichment compared to blank **(left**). FC = fold change.

**Figure 3 nutrients-15-05077-f003:**
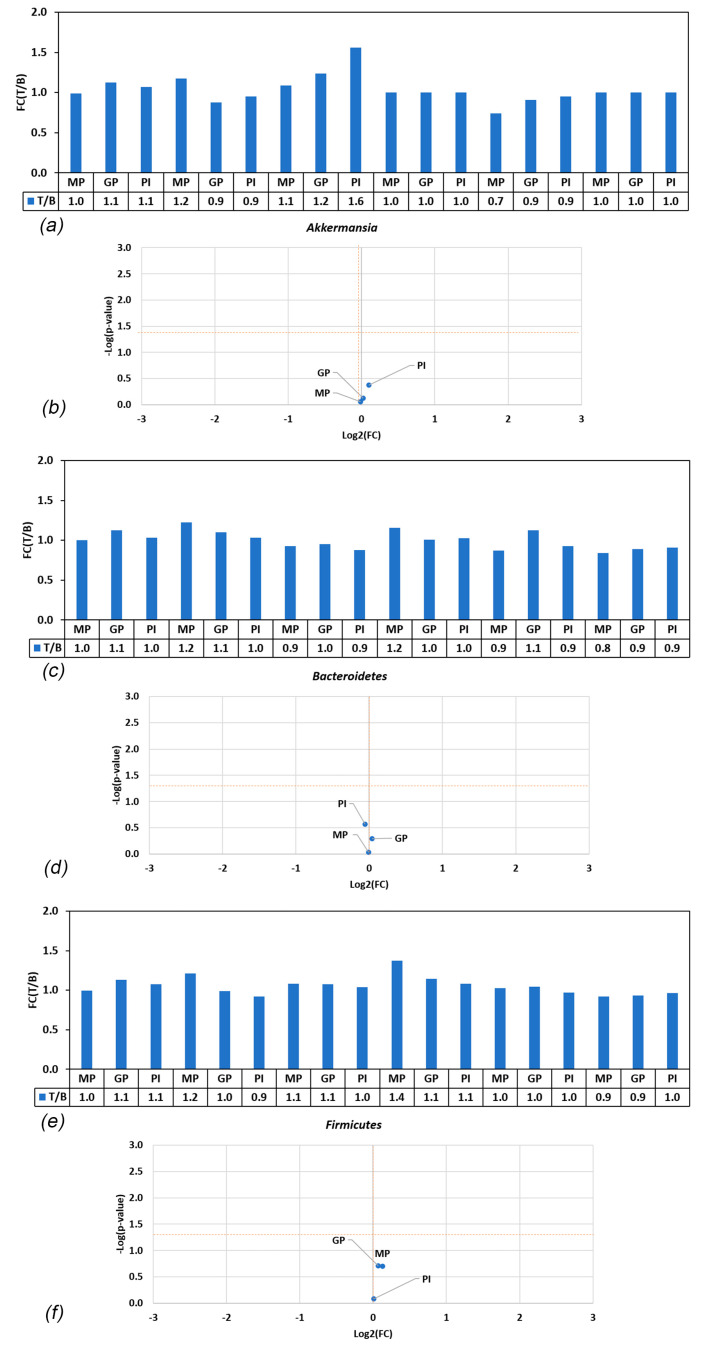
(**a**,**c**,**e**) Ratios of *Akkermansia muciniphila*, *Bacterroidetes* spp., and *Firmicutes* abundances in treatment versus blank (T/B) in the various conditions. Values > 1 indicate enrichment in treatment; values < 1 indicate enrichment in blank. FC = fold change; T = treatment; B = blank; (**b**,**d**,**f**) Volcano plot showing difference in abundance of *A. muciniphila*, *Bacteroidetes* spp., and *Firmicutes* in blank and treatment. Statistical significance (−log(*p*-value)) is plotted in function of fold change (treatment/blank), classifying treatments into one of four categories: non-significantly lower enrichment than blank control (**bottom left**), significantly lower enrichment than blank control (**top left**), non-significantly higher enrichment than blank control (**bottom right**), and significantly higher enrichment than blank control (**top right**). The horizontal orange line indicates the level above which an effect is statistically significant compared to the blank across the six donors. The vertical orange line marks the separation between stronger enrichment versus blank control (**right**) or lower enrichment compared to blank (**left**). FC = fold change.

## Data Availability

The data presented in this study are available on request from the corresponding author. The data are not publicly available due to privacy and ethical restrictions.

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
