# Peer review of "Evidence of the Beneficial Impact of Three Probiotic-Based Food Supplements on the Composition and Metabolic Activity of the Intestinal Microbiota in Healthy Individuals: An Ex Vivo Study"

_nutrients, 2023, doi:10.3390/nu15245077_

Round 1

Reviewer 1 Report

Comments and Suggestions for Authors

The manuscript is well-written and address a relevant and current topic that is of interest to a wide audience of researchers. However, I have identified a few areas that require attention and improvement. Please find the specific points below:

1. The introduction needs more focus on the specific objectives of the investigation. While the authors used the microbiota from healthy donors, a significant portion of the introduction discusses the microbiota-disease relationship. To improve the introduction, the authors should delve into the metabolism of probiotics in the large intestine, provide details on the characteristics of the strains used, the rationale behind their selection, and potential effects of the probiotic formula components on the intestine. Additionally, updating references to include more recent literature will strengthen the introduction.

2. The discussion lacks consideration of the potential effects of additional components in the probiotic formulas (e.g., lactase, bromelain, extracts of chamomile, fennel, mint, lavender flower extract, glutamine, and omega-3 fatty acids) on the composition of the intestinal microbiota and the production of metabolites. It remains unclear whether these components were considered non-impactful or if their potential impact was overlooked by the authors. A commentary on this topic in the discussion section would help clarify any uncertainties for readers.

3. The conclusion appears to extrapolate beyond the obtained results by mentioning "dysbiosis." It is crucial to align the conclusion with the specific findings of the study and avoid making assertions that extend beyond the scope of the research.

Author Response

EVIDENCE OF THE BENEFICIAL IMPACT OF THREE PROBIOTIC-BASED FOOD
SUPPLEMENTS ON THE COMPOSITION AND METABOLIC ACTIVITY OF THE INTESTINAL MICROBIOTA IN HEALTHY INDIVIDUALS. AN EX VIVO STUDY

nutrients-2720744

Dear Editor, dear reviewer

We thank very much to the reviewers for their constructive criticism and suggestions for changes that undoubtedly have served to improve the quality of this manuscript. We are enclosing a revised manuscript, with the changes marked with the track change tool of Microsoft Word, following all the reviewer’s recommendations. In addition, in the present document, answers to all queries and the subsequent changes, undertaken in the manuscript, are detailed.

Best regards,

The Authors

Authors´ Comments to Reviewers:

Reviewer 1:

  1. The introduction needs more focus on the specific objectives of the investigation. While the authors used the microbiota from healthy donors, a significant portion of the introduction discusses the microbiota-disease relationship. To improve the introduction, the authors should delve into the metabolism of probiotics in the large intestine, provide details on the characteristics of the strains used, the rationale behind their selection, and potential effects of the probiotic formula components on the intestine. Additionally, updating references to include more recent literature will strengthen the introduction.

We appreciate the reviewer's insight, which identifies a deficiency in the introduction regarding crucial aspects related to the subject under studyo. Implementing these suggestions will notably enhance the article's quality. In response to the reviewer's guidance, the revised introduction now emphasizes the research objectives and delves into the advantageous effects of the intestinal microbiota on both general health and the individual's gut health, in  the rationale behind the selection of specific strains, as well as information about the bioactive ingredients that accompany the strains in two of the selected products, drawing upon recent scientific literature

It now reads:

“… The importance of the gut microbiome lies in its intimate relationship with the individual’s health, both at the level of the intestinal mucosa and the immune system. Interactions within the host-gut microbiota promote the functional and structural maturation of the gastrointestinal tract, acting mainly on surface maturation and peristalsis, maintaining the integrity of the intestinal epithelial barrier by maintaining cell-cell junctions, and promoting epithelial repair after injury. At the same time, the microbiota provides a physical barrier against incoming pathogens through competitive exclusion by occupying binding sites, consuming nutrient sources, and producing antimicrobial substances, and also by stimulating the host to produce various antimicrobial compounds. Additionally, the metabolic processes of the intestinal microbiota benefit the host and participate in acquiring nutrients, producing essential human metabolites such as vitamins, and processing xenobiotics. The gut microbiota provides capabilities for the fermentation of nondigestible substrates such as dietary fibers and endogenous intestinal mucus. This fermentation supports the growth of short-chain fatty acids (SCFA)-producing bacteria, which are the main source of energy for human colonocytes. Fermentation also supports the grown of microorganisms that facilitate apoptosis in colon cancer cells and that activate intestinal glucose and energy homeostasis [5-7]. …

“… The direct or indirect metabolic action of probiotics includes catalytic pathways for the metabolism of complex carbohydrates that produce the SCFAs. Probiotics also modulate cytokines related to inflammation and have an influence on anti-proliferative lipids, which represent an essential energy source for gastrointestinal epithelial cells, microbial-derived bioactive metabolites such as vitamins K and B, hormones, and neurological signaling molecules, among others [7,13,14]. …

“… Lactobacilli, along with Bifidobacterium species, are historically considered the most common probiotics that, in controlled studies in humans, have demonstrated health benefits. The yeast Saccharomyces boulardii and some species of Escherichia coli and Bacillus are also used. Newcomers to the probiotic ranks include Clostridium butyricum, recently approved as a novel food in the European Union (World Gastroenterology Organization Global Guidelines for Probiotics and Prebiotics, February 2023; www.worldgastroenterology.org). In this context, only a few studies have explored how probiotics affect the metabolism and composition of a healthy intestinal microbiota to maintain a balanced environment. Most studies mainly focus on alleviating symptoms associated with disorders caused directly or indirectly by an imbalance in the gut microbiota. In fact, to improve their effectiveness against symptoms related to these disorders, commercial probiotics are often formulated with a combination of multiple probiotic strains. Alternatively, they might contain a single strain of probiotics or a mixture, combined with natural bioactive agents such as essential oils, coumarins, tannins, flavonoids, phytosterols, amino acids, omega-3 fatty acids, or proteolytic enzymes. These components help relieve symptoms such as abdominal pain, bloating, irregular bowel movements, reflux, and heartburn. It is important to note that these ingredients not only influence intestinal cells by modulating inflammation, redox state, pain perception, and immune responses, but they also potentially affect the balance and activity of the intestinal microbiota. [22-26]…”

“ …Therefore, the aim of this ex vivo study was to evaluate how probiotic-based food supplements containing Lactobacillus and Bifidobacterium strains impact the composition and metabolic activity of the intestinal microbiota [12]. Three probiotic-based food supplements were selected due to their success in treating intestinal dysfunction: one with Bifidobacterium longum ES1 combined with lavender extract, glutamine, and omega-3 fatty acids for symptoms such as abdominal discomfort; another with Lactobacillus acidophilus NCFM® with ingredients such as lactase, bromelain, chamomile, fennel, and mint for acid reflux conditions; and a third combining L. acidophilus NCFM®, Lactobacillus paracasei Lpc-37™, Bifidobacterium lactis Bi-07™, and Bifidobacterium lactis Bl-04™ to address intestinal stress….”

  1. The discussion lacks consideration of the potential effects of additional components in the probiotic formulas (e.g., lactase, bromelain, extracts of chamomile, fennel, mint, lavender flower extract, glutamine, and omega-3 fatty acids) on the composition of the intestinal microbiota and the production of metabolites. It remains unclear whether these components were considered non-impactful or if their potential impact was overlooked by the authors. A commentary on this topic in the discussion section would help clarify any uncertainties for readers.

Thank you to the reviewer for the observation. Indeed, the potential impact of the bioactive ingredients complementing the probiotic strains had not been addressed in the discussion. The primary reason for this omission was to avoid shifting the readers' focus from the probiotic strains to these natural compounds. Notably, the two supplements enriched with bioactive principles did not demonstrate significant effects on the intestinal microbiota of the model employed. Following the reviewer's guidance, this aspect has now been clarified within the discussion.

It now reads:

“…The probiotic strains B. longum ES1 and L. acidophilus NCFM® employed in this study were accompanied by bioactive ingredients. In contrast to findings reported by other authors who observed a beneficial modulation of intestinal microbiota with the use of, for instance, lavender extract, fennel, chamomile or omega 3 fatty acids [22-26], our results indicate that in the two supplemented formulations, PI and GP, the additional components did not have a significant impact on the composition or metabolism of the intestinal microbiota. Lastly, the results support the pre-existing scientific evidence that the benefits increase if more than one probiotic, such as lactobacilli, is combined with bifidobacteria [65,66]…”

  1. The conclusion appears to extrapolate beyond the obtained results by mentioning "dysbiosis." It is crucial to align the conclusion with the specific findings of the study and avoid making assertions that extend beyond the scope of the research.

We thank the reviewer this comment. The authors agree with the reviewer's suggestion and, following his indications, the conclusions have been rewritten to align them with the results found.

It now reads:

“…In conclusion, the results of our study showed that overall, the impact of the tested probiotic agents on the fermentation process of the healthy human microbiome is strain dependent. The combination of the probiotic strains L. acidophilus NCFM®, L. paracasei Lpc-37TM, B. lactis Bi-07TM and B. lactis BI-04TM increased the levels of Bifidobacterium, Lactobacillus and butyrate and reduced the production of BCFA and NH4+ (p<0.05 in all cases), highlighting the importance of this combination of strains for future research in this area, human health, and clinical practice. There was a significant effect on the intestinal microbiome of the healthy individuals, observing changes in metabolic activity and microbial abundance, which could constitute a reduction in disease risk. We can thereby postulate that the use of probiotics could be a powerful preventive approach”

Reviewer 2 Report

Comments and Suggestions for Authors

In the manuscript submitted to me for review entitled "Evidence of the beneficial impact of three probiotic-based food supplements on the composition and metabolic activity of the intestinal microbiota in healthy individuals. An ex vivo study the authors María Carmen Sánchez, Ana Herráiz, Sindy Tigre, Arancha Llama-Palacios, Marta Hernádez, MJ. Ciudad and Luis Collado study the ex vivo effect on the composition and metabolic activity of the intestinal microbiota of three probiotic-based nutritional supplements containing respectively 1) Bifidobacterium longum ES1, 2) Lactobacillus acidophilus NCFM®, and 3) the combination of L. acidophilus NCFM®, Lactobacillus paracasei Lpc-37™, Bifidobacterium lactis Bi-07™ and Bifidobacterium lactis Bl-04™.

The results of the study bring extremely important information for the future proper selection of probiotic strains to be taken as a food supplement to improve both the activity of the digestive system and contribute to the lower probability of developing other health problems with age.

To support their research, the authors used 56 references that cover research for more than three decades of research, including more recent data from the last 5 years - 14 references (1/4 of the total number of references).

The research was conducted and described in an extremely consistent and detailed manner. The results are presented using 3 figures in the main text of the manuscript and 8 supplementary figures. I have no objections to the authors regarding the way the research was conducted, the methods used and the statistical processing of the results. I have only three suggestions for improving individual parts of the manuscript.

1. In my opinion, at least some of the additional figures can be inserted into the manuscript itself. All eight figures bring interesting information from the obtained results and I think it will benefit the manuscript if at least some of them are included in it.

2. In my opinion, about the first half of the discussion pretty much repeats the results section. I think it could be rewritten a bit and results already described in the results here could be dropped or shortened a bit more.

3. Some of the references do not list all the authors (Nos. 3, 4, 9, 12, 14, 16, 19, 24, 26, 29, 33, 34, 35, 36, 37, 43, 45, 46, 47, 48, 50, 51, 52, 53, 55 and 56). Personally, when I read an article, I prefer the references to be fully written, instead of having to search for some of the authors. I think it would be helpful to your readers if all authors in all references are listed.

Author Response

EVIDENCE OF THE BENEFICIAL IMPACT OF THREE PROBIOTIC-BASED FOOD
SUPPLEMENTS ON THE COMPOSITION AND METABOLIC ACTIVITY OF THE INTESTINAL MICROBIOTA IN HEALTHY INDIVIDUALS. AN EX VIVO STUDY

nutrients-2720744

Dear Editor, dear Reviewer

We thank very much to the reviewers for their constructive criticism and suggestions for changes that undoubtedly have served to improve the quality of this manuscript. We are enclosing a revised manuscript, with the changes marked with the track change tool of Microsoft Word, following all the reviewer’s recommendations. In addition, in the present document, answers to all queries and the subsequent changes, undertaken in the manuscript, are detailed.

Best regards,

The Authors

Authors´ Comments to Reviewers:

Reviewer 2:

  1. In my opinion, at least some of the additional figures can be inserted into the manuscript itself. All eight figures bring interesting information from the obtained results and I think it will benefit the manuscript if at least some of them are included in it.

We appreciate the reviewer's observation. The authors concur with the reviewer's perspective regarding the valuable insights offered by the supplementary figures, which contain pertinent information for the study. Ideally, we would have included all figures in the main text. However, due to their size and the extensive range of variables studied involving metabolites and bacterial species, we opted to present three figures summarizing the results. We hope that this decision does not pose a barrier to the publication of the article. Should the reviewer perceive this as an issue, we are open to discussing it with the journal's editors concerning constraints related to page limitations and the number of figures allowed.

  1. In my opinion, about the first half of the discussion pretty much repeats the results section. I think it could be rewritten a bit and results already described in the results here could be dropped or shortened a bit more.

Thank you for the reviewer's feedback. It was noted that a summary of the obtained results was utilized in the discussion, which could potentially introduce redundancy. In response to this observation, the paragraph summarizing the results has been omitted from the discussion to enhance the section's readability, aligning with the reviewer's suggestion.

  1. Some of the references do not list all the authors (Nos. 3, 4, 9, 12, 14, 16, 19, 24, 26, 29, 33, 34, 35, 36, 37, 43, 45, 46, 47, 48, 50, 51, 52, 53, 55 and 56). Personally, when I read an article, I prefer the references to be fully written, instead of having to search for some of the authors. I think it would be helpful to your readers if all authors in all references are listed.

We appreciate the reviewer's comment, and the authors wholeheartedly concur. As per the journal's guidelines, references are formatted in compliance with specific rules, necessitating the listing of only the initial 10 authors in multi-author articles. Unfortunately, we regret our inability to address this matter.

Reviewer 3 Report

Comments and Suggestions for Authors

The topic of intestinal microbiota and its effect on health is timely and it is largely vague to many clinicans. The Authors present a research aimed to assess the effect of three probiotic-based food supplements on the composition and meatbolic functions of intestinal microbiota in ex vivo study.

Overall the manuscript is logically organized, well edited and well written in English. The aim of the study is well described and study enpoints clear. The results are well described and figures readable. However, several elements regarding the research methods raise my doubts and require clarification. Below are my suggestions:

Major suggestions:

1. Can you explain (in the Introduction) the choice of these specific strains?

2. Why did the authors compare probiotics supplements that contain/or do not contain other abioactive ingredients? What is the potential impact of additional bioactive substances eg. EPA/DHA on the results?

3. Could Authors provide information about probiotics doses? If supplements provided comparable doses of probiotic bacteria?

4. On what basis were the 6 donors selected? Is such a small number of samples representative?

5. Conslusion: Please provide one-two additional sentences whis highlight the importance of the study for future reasearches in this area, human health and clinical practice.

Minor suggestions:

page 2, line 67-> Please, change type II diabetes to "type 2 diabetes"

Refferences -> double numeration

Author Response

EVIDENCE OF THE BENEFICIAL IMPACT OF THREE PROBIOTIC-BASED FOOD
SUPPLEMENTS ON THE COMPOSITION AND METABOLIC ACTIVITY OF THE INTESTINAL MICROBIOTA IN HEALTHY INDIVIDUALS. AN EX VIVO STUDY

nutrients-2720744

Dear Editor, dear Reviewer

We thank very much to the reviewers for their constructive criticism and suggestions for changes that undoubtedly have served to improve the quality of this manuscript. We are enclosing a revised manuscript, with the changes marked with the track change tool of Microsoft Word, following all the reviewer’s recommendations. In addition, in the present document, answers to all queries and the subsequent changes, undertaken in the manuscript, are detailed.

Best regards,

The Authors

Authors´ Comments to Reviewers:

Reviewer 3:

  1. Can you explain (in the Introduction) the choice of these specific strains?

Thank you for your insightful observation. It's true that the introduction lacked references to the selected strains and the rationale behind their selection. In response to your valuable suggestion, we have now provided a clear rationale for our choice within the introduction.

It now reads:

“… Lactobacilli, along with Bifidobacterium species, are historically considered the most common probiotics that, in controlled studies in humans, have demonstrated health benefits. The yeast Saccharomyces boulardii and some species of Escherichia coli and Bacillus are also used. Newcomers to the probiotic ranks include Clostridium butyricum, recently approved as a novel food in the European Union (World Gastroenterology Organization Global Guidelines for Probiotics and Prebiotics, February 2023; www.worldgastroenterology.org). In this context, only a few studies have explored how probiotics affect the metabolism and composition of a healthy intestinal microbiota to maintain a balanced environment. Most studies mainly focus on alleviating symptoms associated with disorders caused directly or indirectly by an imbalance in the gut microbiota. In fact, to improve their effectiveness against symptoms related to these disorders, commercial probiotics are often formulated with a combination of multiple probiotic strains. Alternatively, they might contain a single strain of probiotics or a mixture, combined with natural bioactive agents such as essential oils, coumarins, tannins, flavonoids, phytosterols, amino acids, omega-3 fatty acids, or proteolytic enzymes. These components help relieve symptoms such as abdominal pain, bloating, irregular bowel movements, reflux, and heartburn. It is important to note that these ingredients not only influence intestinal cells by modulating inflammation, redox state, pain perception, and immune responses, but they also potentially affect the balance and activity of the intestinal microbiota. [22-26].

Therefore, the aim of this ex vivo study was to evaluate how probiotic-based food supplements containing Lactobacillus and Bifidobacterium strains impact the composition and metabolic activity of the intestinal microbiota [12]. …”

  1. Why did the authors compare probiotics supplements that contain/or do not contain other bioactive ingredients? What is the potential impact of additional bioactive substances eg. EPA/DHA on the results?

The authors appreciate the reviewer's point about insufficient clarification in the introduction. Based on the reviewer's guidance, the introduction was revised to lay out the rationale behind the supplements and explore the potential effects of their bioactive ingredients on the gut microbiome. Furthermore, the discussion now summarizes the conclusion derived from the results related to these bioactive ingredients accompanying the probiotic strains.

It now reads:

 “… Lactobacilli, along with Bifidobacterium species, are historically considered the most common probiotics that, in controlled studies in humans, have demonstrated health benefits. The yeast Saccharomyces boulardii and some species of Escherichia coli and Bacillus are also used. Newcomers to the probiotic ranks include Clostridium butyricum, recently approved as a novel food in the European Union (World Gastroenterology Organization Global Guidelines for Probiotics and Prebiotics, February 2023; www.worldgastroenterology.org). In this context, only a few studies have explored how probiotics affect the metabolism and composition of a healthy intestinal microbiota to maintain a balanced environment. Most studies mainly focus on alleviating symptoms associated with disorders caused directly or indirectly by an imbalance in the gut microbiota. In fact, to improve their effectiveness against symptoms related to these disorders, commercial probiotics are often formulated with a combination of multiple probiotic strains. Alternatively, they might contain a single strain of probiotics or a mixture, combined with natural bioactive agents such as essential oils, coumarins, tannins, flavonoids, phytosterols, amino acids, omega-3 fatty acids, or proteolytic enzymes. These components help relieve symptoms such as abdominal pain, bloating, irregular bowel movements, reflux, and heartburn. It is important to note that these ingredients not only influence intestinal cells by modulating inflammation, redox state, pain perception, and immune responses, but they also potentially affect the balance and activity of the intestinal microbiota. [22-26].

Therefore, the aim of this ex vivo study was to evaluate how probiotic-based food supplements containing Lactobacillus and Bifidobacterium strains impact the composition and metabolic activity of the intestinal microbiota [12]. Three probiotic-based food supplements were selected due to their success in treating intestinal dysfunction: one with Bifidobacterium longum ES1 combined with lavender extract, glutamine, and omega-3 fatty acids for symptoms such as abdominal discomfort; another with Lactobacillus acidophilus NCFM® with ingredients such as lactase, bromelain, chamomile, fennel, and mint for acid reflux conditions; and a third combining L. acidophilus NCFM®, Lactobacillus paracasei Lpc-37™, Bifidobacterium lactis Bi-07™, and Bifidobacterium lactis Bl-04™ to address intestinal stress…”

  1. Could Authors provide information about probiotics doses? If supplements provided comparable doses of probiotic bacteria?

 Thank you for your observation. Each commercial product contains varying doses of the probiotic strains: MP (1.5 E+7 CFU/g), GP (1.3 E+8 CFU/g), and PI (3.3 E+8 CFU/g). It's important to note that our study aimed to assess the impact of these probiotic strains specifically at the intestinal level, rather than focusing on the overall commercial product. As detailed in the materials and methods section, we standardized our experimentation using a final concentration of 1E+07 CFU/mL across all three supplements.

  1. On what basis were the 6 donors selected? Is such a small number of samples representative?

 Thank you for your observation. To ensure a representative sample, we selected a group consisting of 3 men and 3 women from the healthy adult population, as outlined in the materials and methods section. This sample size was deemed sufficient to conduct a robust statistical analysis with significant power.

  1. Conclusion: Please provide one-two additional sentences which highlight the importance of the study for future researchers in this area, human health and clinical practice.

Thank you very much for your appreciation. Following the reviewer's guidance, the conclusion has been rewritten to highlight the importance of the study results.

It now reads:

“In conclusion, the results of our study showed that overall, the impact of the tested probiotic agents on the fermentation process of the healthy human microbiome is strain dependent. The combination of the probiotic strains L. acidophilus NCFM®, L. paracasei Lpc-37TM, B. lactis Bi-07TM and B. lactis BI-04TM increased the levels of Bifidobacterium, Lactobacillus and butyrate and reduced the production of BCFA and NH4+ (p<0.05 in all cases), highlighting the importance of this combination of strains for future research in this area, human health, and clinical practice. There was a significant effect on the intestinal microbiome of the healthy individuals, observing changes in metabolic activity and microbial abundance, which could constitute a reduction in disease risk. We can thereby postulate that the use of probiotics could be a powerful preventive approach.”

6. Page 2, line 67-> Please, change type II diabetes to "type 2 diabetes"

 Thank you very much for your observation. Following the reviewer's instructions, the term has been corrected

 7. References -> double numeration

Following the reviewer's instructions, the references have been revised and double numbering removed.

Round 2

Reviewer 3 Report

Comments and Suggestions for Authors

I thank the Authors for considering and responding to all comments and suggestions. The introduced corrections allow us to conclude that the manuscript can be published in its current form.